# Indoor Radon Research in the Asia-Pacific Region

**Miroslaw Janik** [1,*] , **Peter Bossew** [2] , **Md. Mahamudul Hasan** [3] and **Giorgia Cinelli** [4]

1   National Institute of Radiological Sciences (QST/NIRS), Chiba 263-8555 , Japan
2   German Federal Office for Radiation Protection (BfS), 10318 Berlin, Germany
3   Department of Environment Systems, Graduate School of Frontier Sciences, The University of Tokyo, Kashiwa Campus, Chiba 277-0882, Japan
4   National Agency for New Technologies, Energy, and Sustainable Economic Development (ENEA), 90139 Palermo, Italy
*   Correspondence: janik.miroslaw@qst.go.jp

**Abstract:** Indoor radon is a major hazard to human health; it is one of the leading causes of lung cancer. Therefore, radon research in Asia has intensified recently due to the growing awareness of the harm that radon poses. An analysis of the collected literature data showed that in Asia–Oceania, some regions have—or are believed to have—little indoor radon problems due to climate and low Rn ground. It can be concluded that countries have their own approaches, techniques, and protocols. Data were not harmonized within each region; however, measurement techniques were compared by regional intercomparison exercises. The situation differs regarding studies on the usability of radon as a tracer or potential predictor of tectonic phenomena, as some countries are in seismically active zones, such as India, Taiwan, China, Japan, etc. India and Taiwan are global leaders in this research, as well as Italy, which is another seismically affected country. We provide an overview of radon-related surveying and research activities conducted in the western, southern, and eastern Asian regions over the past few years. Additionally, we observed that the number of indoor radon measurements per million inhabitants increases as the human development index (HDI) increases.

**Keywords:** radon; regulation; Asia; HDI





## 1. Introduction

Radon (Rn), a radioactive gas naturally emanating from Earth's crust, accounts for about one-half of the effective dose of ionizing radiation received by humans. The most relevant isotopes are $^{222}$Rn, hereafter referred to as 'radon', which is a decay product of the $^{238}$U decay chain, and $^{220}$Rn, hereafter referred to as 'thoron', which is a decay product of the $^{232}$Th decay chain. Radon measurements in various uranium and non-uranium mines began in the early 20th century. Based on this, it was soon assumed that radon could be responsible for lung cancer among underground miners [1]. Two studies with quantitative analyses among the US and Czechoslovakian miners concluded that the lung cancer risk increased monotonically with cumulative exposure to radon progeny [2,3]. Motivated by these studies, further efforts were made to develop more reliable methods for monitoring radon progeny in mines. It was found that radon progeny ranges from several to thousands of kBq m$^{-3}$ [4].

The results of the first set of indoor radon measurements in Sweden indicated high levels of radon in several houses built with radium-rich alum shale concrete [5]. Since then, large indoor radon surveys have been carried out in several countries, for example, the USA, many European countries (e.g., the UK and Czech Republic), and Japan. In some countries, such studies are currently being conducted, e.g., in India and China.

General findings from numerous epidemiological studies, based on data from miners and the general population, support the conclusion that, following cigarette smoking, prolonged residential radon exposure is one of the leading causes of lung cancer in the general population [6–10].

However, contrary to these findings, there are observations that do not confirm that statement. A series of cancer mortality studies near radon hot springs were conducted through the collaborative efforts scattered throughout different Japanese cities. At present, there is no definite evidence indicating an increase in cancer mortality in the Misasa radon hot spring area. Moreover, lower risks were found for stomach cancers in both radon and non-radon hot springs [11–13]. In Ramsar, Iran, inhabitants are exposed to levels of natural radiation that are about 150 times higher than the average global dose rate; indoor radon concentrations are up to 3700 Bq m$^{-3}$. It was concluded that the lung cancer rate may show a negative correlation with the natural radon concentration [14].

In addition, the results of a meta-analysis involving thirty-two case-control studies and two ecological radon studies on lung cancer, focusing on radon concentrations below 1000 Bq m$^{-3}$, do not support the finding that radon may be a cause of a statistically significant increase in the incidence of lung cancer [15]. The discussion about the effects of low radon exposure as well as low-level ionizing radiation is ongoing [16,17].

In any case, internationally, indoor radon is recognized as a health issue and a preventable risk factor that can be handled through effective national policies and regulations [18]. Consequently, due to the potential effect of radon on human health, it has been the subject of many studies worldwide. Currently, work is being conducted on both radon monitoring and epidemiology. Radon monitoring and control strategies focus on indoor and underground spaces where radon can accumulate due to limited ventilation.

On the other hand, due to their physical properties and dynamic behavior, radon and its progeny can be employed as tracers in geohazard and atmospheric studies, for example, in the prediction of earthquakes and volcanic events, as well as in the analysis of mass transport and mixing processes.

It can be concluded that radon should be treated by its negative and positive aspects, i.e., not only as a health threat but also as a useful research tool, as summarized in Table 1.

**Table 1.** Positive and negative aspects of radon.

| Positive Aspects | Negative Aspects |
|---|---|
| • Earthquake prediction [19–21]<br>• Atmospheric studies, climate research [22–26]<br>• Volcanic surveillance [27]<br>• Lunar science [28]<br>• Mineral exploration [29,30]<br>• Geothermal energy prediction [31,32]<br>• Balneotherapy in radon spas (USA, Japan, and Europe) [33–35]<br>• Search for organic pollutants in the ground [36] | • Health effects, i.e., cancer and leukemia [37,38]<br>• Contributor to radiation doses in NORM, e.g., mineral factory [39]<br>• Background in laboratories, e.g., Super-Kamiokande neutrino observation laboratory [40] |

In this paper, Section 2 provides a summary of radon policy and regulations in Asian and Oceanic countries. We also compare these regulations with those from other regions around the world, especially the EU and North America. In Section 3, we present an overview of radon surveys by country, citing references that we could find in the literature. In Section 4, we discuss issues mainly related to QA, as relevant from the evaluated literature. Section 5 addresses new technical and scientific development, with selected examples from countries in the region. In Section 6, we present a statistical evaluation of the survey density, discuss the results, and propose our conclusions.

## 2. Radon Regulations

In Europe, the EURATOM Basic Safety Standards, which were published in 2013, include binding requirements (to be implemented into national legislation) for protecting against indoor radon exposure at work, home, and in the manufacturing and use of building materials [41]. Therefore, many radon projects are underway in Europe. One of the achievements of European radon research is the development of the European Indoor Radon Map, which is part of the European Atlas of Natural Radiation, created by the Joint Research

Centre of the European Commission [42]. Other achievements include the EU-funded Metro Radon project (http://metroradon.eu, accessed on 1 February 2023) which focuses on QA, from radon measurement to the delineation of radon priority areas; moreover, the traceRadon project (http://traceradon-empir.eu, accessed on 1 February 2023) provides a necessary scientific base for measuring atmospheric radon activity concentrations and radon fluxes [25]. According to the recommendations of the WHO [43] and the European Commission [41], a reference level (RL) of 300 Bq m$^{-3}$ has been established, and the annual average indoor radon concentration should not exceed this value. National RLs vary across different countries due to the variations in regional levels of indoor radon, which usually range from 100 to 300 Bq m$^{-3}$.

In the USA and Canada, the action level, i.e., a threshold for recommending or requiring mitigation of exposure to harmful elements, is set at 148 Bq m$^{-3}$ (4 pCi/L) and 200 Bq m$^{-3}$, respectively [44,45], with no distinction between existing and new dwellings. In Asia and Oceania, the countries with radon regulations are limited, according to the WHO Radon Database [46] and other collected references; see Table 2.

**Table 2.** Reference levels (RL) in dwellings, mitigation, and prevention actions. AM—national mean indoor Rn concentration (Bq m$^{-3}$).

| Country | RL | AM [a] | Mitigation | Prevention | Reference |
|---------|-----|--------|-----------|-----------|-----------|
| Australia | 200 | 10.5 | No | No | [46] |
| Bahrain | 300 | - [b] | - | - | [46] |
| China | 300 | 40 | - | - | [46,47] |
| Georgia | 300 | - | - | - | [46] |
| Mongolia | 100 | - | - | - | [46] |
| Turkey | 400 | 81 | Yes | No | [46] |
| Turkmenistan | 200 | - | - | NA | [48] |
| Korea | 200 | 62 | - | - | [49,50] |
| Israel | 200 | 31 | - | - | [51] |
| Kazakhstan | 100 (new), 200 (old) | - | - | - | [52] |

[a] value from the latest survey, [b] not available.

There is no country in Asia or Oceania with a national radon risk communication strategy; however, in some of these countries, strategies against pollutants other than radon exist, as listed in Table 3.

**Table 3.** Radon communication and linkage to other national strategies (source: WHO).

| Country | Cancer Control Strategy | Lung Cancer Reporting/ Screening Strategy | Indoor Air Quality Strategy | Energy Conservation Strategy |
|---------|------------------------|-------------------------------------------|----------------------------|------------------------------|
| Australia | X | X | | |
| China | X | X | X | X |
| Iran | X | | | |
| Mongolia | X | | | |
| Turkey | | | X | X |

## 3. Radon Surveys

Surveys can be classified according to their design characteristics. Among the criteria are their objectives (assessments of geographical or demographic means, i.e., means per area unit or per person living in an area, which are generally different), their coverage (which part of a country does a survey cover, distinguishing between local, regional, and national surveys), and the degree of representativeness. The latter indicates whether derived statistics, such as the empirical mean, can be assumed to coincide with the respective true value of the sampled quantity.

In this article, we do not make a distinction between surveys according to their design characteristics, in particular, whether they are regional or national surveys. Details, where

available, are given in the relevant Section 3.1. See Section 4.3 for further information on this important subject.

Based on 2019 WHO data ( https://www.who.int/data/gho/data/themes/topics/topic-details/GHO/gho-phe-radon-database, accessed on 1 February 2023) only Australia, China, Turkey, and Syria conducted national radon surveys. However, as we will discuss later, there are (or have been) large-scale radon studies in several countries, namely India (ongoing), Iran, Israel, Japan, Korea, New Zealand, and the Philippines. Russia is not discussed in this paper since it is usually categorized with Europe. On the other hand, Turkey, Armenia, Georgia and Azerbaijan are discussed here, although they are often counted with Europe.

A summary of radon levels is presented in Table 4. The countries are presented by region, i.e., central Asia (I), eastern Asia (II), southeastern Asia (III), southern Asia (IV), and western Asia (V), according to the methodology introduced by the United Nations [53]. The list might not be complete because sometimes survey results are published in literature that is difficult to access. Therefore, it may also be that in countries not mentioned here, radon data exist. The first meta-survey on radon in Asia and Oceania was presented in 2019 at the 16th AOGS conference in Singapore [54]. New data have been added since then.

The mean values shown in Table 4 are weighted average values (*WAM*) in case that in a country several surveys have been performed. The weighted average was calculated by multiplying the average radon concentration by the number of measurements for a given survey, then dividing by the total number of measurements using following equation: $WAM = sum(AM_i * n_i)/sum(n_i)$, where $AM_i$ is the average radon concentration from the $i-th$ survey and $n$ is the number of measurement points during $i-th$ survey.

We should note that the averages given can only be used to estimate true geographical or demographic measures if the study was representative for the purpose intended. Empirical averages that are calculated from scattered data should not be interpreted as valid averages at the national level. In particular, the demographic average, which is the average of a demographically representative study, is generally not equal to the geographic average. It should also be noted that the regional survey averages are not representative of the national average.

**Table 4.** Average and maximum indoor radon concentrations (Bq m$^{-3}$) with the number of measurement points.

| Country | ISO Code | Rn Survey Average (WAM) | Rn Max | Subregion | Number of Measurement Points |
|---|---|---|---|---|---|
| - | - | Bq m$^{-3}$ | Bq m$^{-3}$ | - | - |
| Afghanistan | AF | 65 | 2064 | IV | 16 |
| Armenia | AM | * | 400 | V | 800 |
| Australia | AU | 12 | 423 | Oceania | 3413 |
| Azerbaijan | AZ | 84 | 1100 | V | 2404 |
| Bangladesh | BD | 113 | 2616 | IV | 308 |
| Brunei | BN | ** | ** | III | 1 |
| China | CN | 37 | 1244 | II | 144,937 |
| Georgia | GE | 114 | 376 | V | 28 |
| Hong Kong | HK | 155 | 700 | II | 1580 |
| India | IN | 32 | 373 | IV | 895 |
| Indonesia | ID | 96 | 1015 | III | 394 |
| Iran | IR | 198 | 3700 | IV | 3194 |
| Iraq | IQ | 38 | 239 | V | 175 |
| Israel | IL | 90 | 200 | V | 45,415 |
| Japan | JP | 18 | 1256 | II | 11,360 |
| Jordan | JO | 70 | 1532 | V | 3904 |
| Kazakhstan | KZ | 114 | 37,650 | I | 246 |
| Korea | KR | 91 | 2810 | II | 11,106 |
| Kuwait | KW | 33 | 595 | V | 1108 |
| Kyrgyzstan | KG | 200 | 1996 | I | 68 |
| Lebanon | LB | 39 | 343 | V | 65 |

**Table 4.** *Cont.*

| Country | ISO Code | Rn Survey Average (WAM) | Rn Max | Subregion | Number of Measurement Points |
|---|---|---|---|---|---|
| - | - | Bq m$^{-3}$ | Bq m$^{-3}$ | - | - |
| Malaysia | MY | 22 | 196 | III | 183 |
| Myanmar | MM | 17 | 84 | III | 65 |
| Nepal | NP | 123 | 2206 | IV | 108 |
| New Zealand | NZ | 21 | 302 | Oceania | 977 |
| Oman | OM | 21 | 39 | V | 9 |
| Pakistan | PK | 40 | 191 | IV | 3041 |
| Palestine | PS | 98 | 984 | V | 88 |
| Papua New Guinea | PG | 13 | 18 | Oceania | 60 |
| Philippines | PH | 21 | 58 | III | 2626 |
| Qatar | QA | 16 | 42 | V | 84 |
| Saudi Arabia | SA | 26 | 195 | V | 2955 |
| Singapore | SG | 15 | 80 | III | 10 |
| Syria | SY | 44 | 524 | V | 1435 |
| Taiwan | TW | 11 | 51 | II | 274 |
| Tajikistan | TJ | 76 | 2000 | I | 70 |
| Thailand | TH | 36 | 405 | III | 1541 |
| Turkey | TR | 81 | 1400 | V | 7293 |
| United Arab Emirates | AE | 40 | 71 | V | 61 |
| Uzbekistan | UZ | 219 | 1050 | I | 25 |
| Vietnam | VN | 79 | 634 | III | 142 |
| Yemen | YE | 43 | 890 | V | 293 |

* Data not available, ** Problem with data, explained in the text.

### 3.1. Afghanistan (IV)

The results of the radon surveys in Afghanistan's dwellings are described in [55]. Indoor radon measurements were carried out in two phases from October 2014 to September 2016, with different measurement periods, i.e., one week and one year, using a detector consisting of a diffusion chamber and a microcontroller for data acquisition. The detector was calibrated and compared with an AlphaGuard device [56]. The radon concentration in the shorter phase ranged from 6 to 120 Bq m$^{-3}$ and 25 to 139 Bq m$^{-3}$ for the basements and caves, respectively. The radon concentration in the second phase was from 33 to 2064 Bq m$^{-3}$. It should be noted that the highest values, more than 2000 Bq m$^{-3}$, were observed on one of the basement floors. Excluding these extreme values, the average concentration was found to be 65 Bq m$^{-3}$.

### 3.2. Armenia (V)

The beginning of radon measurements dates back to the 1990s, although the results are not available. The presented materials at a regional workshop organized by IAEA show that a radon program is being conducted in Armenia; a total of 800 alpha track detectors were deployed as part of it in 2010 and 2011 [57]. It was found that, in 147 cases, radon concentration was greater than 200 Bq m$^{-3}$.

In 2011, with support from the IAEA program, a survey on the distribution of radon gas concentration within the territory of Armenia was performed. In 2019, as part of the IAEA RER 9153 program, a survey on the distribution of radon gas concentrations was conducted, focusing on schools and houses in Yerevan, the capital of Armenia; however, the results have not been published as yet [58].

### 3.3. Australia (Oceania)

To estimate the annual average concentration of radon in Australian dwellings and calculate the average annual dose equivalents to the Australian population resulting from radon exposure, a nationwide survey was conducted. The details of this survey are described in the report issued by the Australian Radiation Laboratory in 1990 [59]. Solid

state nuclear track detectors (SSNTDs) were exposed randomly in 3400 homes, which accounted for approximately 1 in 1400 occupied dwellings, to measure radon exposure for 12 months. As a result, the average radon concentration was determined to be at the level of 12 Bq m$^{-3}$.

### 3.4. Azerbaijan (V)

The results of the first indoor radon study in Azerbaijan were presented by Hoffmann et al. [60]. The goal was to create a map with radon distribution and elevated concentrations. Passive detectors were randomly distributed across the country, mostly in apartment buildings. The results from the 2404 homes showed a log-normal distribution with a median of 58 Bq m$^{-3}$ and an average of 84 Bq m$^{-3}$.

### 3.5. Bangladesh (IV)

In Bangladesh, indoor radon concentrations are mostly relatively low; however, among the investigated dwellings, mud-type houses present higher indoor radon concentrations than other types of dwellings (>1000 Bq m$^{-3}$). As part of the conducted measurements, thoron concentration surveys were also performed [61,62].

### 3.6. Brunei (III)

Thus far, one paper has been published on indoor radon measurements in Brunei. The experiment was carried out on the ground and first floors of the Physics Department, Faculty of Science, Universiti Brunei Darussalam. The active measurement system consists of an air filter pump, ZnS detector, and photomultiplier tube. It was found that the concentration of radon in rooms located on the ground floor was 0.39 Bq m$^{-3}$, which was about three times higher than on the first floor [63]. (As a comment, this value is certainly wrong).

### 3.7. China, Hong Kong, Taiwan (II)

In China, taking into account historical data from the years 1980 to 2019, a retrospective study of radon in residential buildings was carried out. A new database was created covering 147 cities with a sample size of 72,295. The average radon concentration derived from these surveys, which were weighted based on sampling size and population for different time periods, ranged from–54 Bq m$^{-3}$. The authors concluded that the average rate of increase in residential radon concentration for 28 Chinese cities was estimated to be 0.80 Bq m$^{-3}$/year in the last 40 years [47].

Another paper from China reviewed 114 surveys and found that the mean concentration of indoor radon for dwellings was 55 Bq m$^{-3}$ [64].

The results from the Yangjiang region, which is recognized as a high-background radiation area (HBRA), showed relatively elevated average radon concentrations of 127 Bq m$^{-3}$ compared to the Chinese average of 37 Bq m$^{-3}$, and more than 1200 Bq m$^{-3}$ for thoron [65].

Indoor Rn concentrations were measured in the main cities of Inner Mongolia; the average indoor Rn concentration was 33 Bq m$^{-3}$ [66].

In Hong Kong, a survey with 1000 detectors showed a mean overall indoor radon concentration of 178 Bq m$^{-3}$ and a standard deviation of 150 Bq m$^{-3}$. The mean radon concentration in dwellings was lower than that in offices and public buildings [67]. In two other studies, the radon concentration in dwellings in Hong Kong was reported to range from 30–155 Bq m$^{-3}$ [68,69].

Indoor radon measurements were conducted in 250 homes across Taiwan using passive cellulose nitrate film in the years 1988 and 1990. The long-term measurements yielded an average indoor concentration of 10 Bq m$^{-3}$ [70]. (As a comment, this value appears too low to be true).

### 3.8. Central Asian Countries (I)

A comprehensive study about radon and thoron measurements at selected former uranium mining and processing sites in the central Asian countries of Kazakhstan, Kyrgyzstan, Uzbekistan, and Tajikistan was presented by Stegnar et al. [71].

### 3.9. Georgia (V)

In total, 15 areas (workplaces and dwellings) were tested for indoor radon concentrations from August 2003 to September 2004 in Tbilisi. The radon concentrations ranged from 19 Bq m$^{-3}$ to 376 Bq m$^{-3}$, with a mean value of 70 Bq m$^{-3}$ [72].

### 3.10. India (IV)

A large-scale surveying program in India is currently underway. It involves the mapping of ambient gamma radiation levels to identify high and low background regions; measuring concentrations of radon, thoron, and their decay products in residential houses for future low-dose epidemiological studies; measuring radon in groundwater for uranium exploration; continuously measuring radon emissions from fault regions for seismic studies; and measuring radionuclide concentrations, radon and thoron emissions from soil, building materials, and TENORM. To execute this national program, a range of cost-effective indigenous equipment has been developed to make the project economical.

So far, based on partial results, the average radon concentration levels have ranged from 1 to 445 Bq m$^{-3}$; the measurements include the HBRA in Kerala, as well as Dasarlapally, which is the proposed uranium site region [73–79].

It should be noted that a literature survey conducted in 2022 showed that in India, where a radon program has been initiated, as previously mentioned, researchers focused on long-term indoor measurements of radon, thoron, and progeny to assess the levels of those radionuclides and calculate inhalation dose [74,80]. The same is being planned for Thailand [81] and Korea [82].

### 3.11. Indonesia (III)

Radon and thoron gases in dwellings on Bali island, Indonesia, ranged from 9 to 48 Bq m$^{-3}$ for radon and below the detection limit of 66 Bq m$^{-3}$ for thoron [83]. However, recent studies on radon concentrations in the HNBRA, located in Mamuju, West Sulawesi, showed a geometric mean of 270 Bq m$^{-3}$ with a range of 90–1644 Bq m$^{-3}$, while Tn concentrations had a geometric mean of 210 Bq m$^{-3}$ with a range of 46–2244 Bq m$^{-3}$ [84,85].

### 3.12. Iran (IV)

The radon level inside 50 dwellings in Shabestar County, Iran, ranged from 4 to 520 Bq m$^{-3}$, with a mean value of 56 Bq m$^{-3}$ [86]. The results of a two-year survey of indoor radon variations in four cities located in north and northwest Iran showed average radon concentrations throughout the year in the range of 124–240 Bq m$^{-3}$. The maximum concentration of 2386 Bq m$^{-3}$ was registered in the winter, while the minimum at a level of 55 Bq m$^{-3}$ in the spring [87].

In Ramsar, a well-known HBRA area, radon was measured in 437 rooms and 16 schools located in high-background and normal-background radiation areas. The mean radon levels in some Ramsar regions were found to be with the mean value of 650 Bq m$^{-3}$ with a maximum value of 3700 Bq m$^{-3}$ [88].

### 3.13. Iraq (V)

In Iraq, a set of indoor radon measurements was carried out from September to November 2011. The results show that the radon concentrations ranged from 39 to 200 Bq m$^{-3}$ [89]. Another study from ten houses showed that the radon concentration range was between 49 and 121 Bq m$^{-3}$ [90].

### 3.14. Israel (V)

In Israel, radon measurements were conducted in 1998 (1800 tests), 2007 (1318 tests), 2008 (1584 tests), and 2011 (1096 tests). In the years 1992–2001 and 1991–2012, a total of 14,100 and 25,000 tests were performed. AM values obtained in different periods ranged between 65 and 94 Bq m$^{-3}$. A recent study involving school-age children across Israel found an average radon concentration of 42 Bq m$^{-3}$ [51,91].

### 3.15. Japan (II)

In Japan, an extensive investigation of indoor radon was performed in three nation-wide surveys [92]. The aim was to obtain the annual average indoor radon level and estimate the population dose. In total, more than 10,000 measurements were performed using passive monitors. In the first surveys performed in the late 1980s and early 1990s, in total, 5700 indoor Rn data points were gathered. The arithmetic mean of the radon level was 21 Bq m$^{-3}$ [93]. In the second campaign organized from 1993 to 1996, a total of 900 measurements were collected. The average annual radon concentration was calculated as 16 Bq m$^{-3}$ [94]. A third survey carried out from 2007 to 2010 covered 3500 dwellings. The average radon concentration with seasonal correction was 14 Bq m$^{-3}$ [95]. It should be noted that the results from the second and third studies were lower than those from the first study. One possible explanation is that radon- and thoron-discriminating detectors were not used during the first study.

In addition to the national study, a number of regional measurements were carried out in dwellings and workplaces.

A radon research study conducted in twenty dwellings in Kumamoto city showed that the average radon concentration was 30 Bq m$^{-3}$, ranging from 13 to 93 Bq m$^{-3}$ [96].

In Fukushima Prefecture, radon levels in temporary housing, apartments, and single-family homes were tested following the Fukushima accident. The measurement results indicated low radon concentrations of 5–9 Bq m$^{-3}$ in all building types [97].

Radon and thoron concentrations were measured for a period of one year in a landmark high-rise building in Tokyo City. The mean annual concentrations were determined to be 16 Bq m$^{-3}$ for radon and 16 Bq m$^{-3}$ for thoron [98].

The latest results from Japan confirm that in some regions of the country, radon and thoron concentrations can be elevated. For example, Furukawa et al. [99] reported higher indoor radon concentrations, i.e., >100 Bq m$^{-3}$, in some ordinary concrete dwellings compared to the Japanese annual average, 16 Bq m$^{-3}$. One of the findings is the observation of an increase in radon concentration during the winter season, reaching a maximum value of 289 Bq m$^{-3}$. The investigation into the origin of this relatively high concentration is planned, as the exact source is currently unknown.

Moreover, three weeks of measurements in Okinawa using the active method also showed elevated radon concentrations exceeding 1000 Bq m$^{-3}$, with an average concentration of about 400 Bq m$^{-3}$ [100]. The same high temporal variability, with occasional very high concentrations >1000 Bq m$^{-3}$, was also observed during the long-term measurements of radon in the basement room of the institute building [101].

### 3.16. Jordan (V)

Several regional studies were conducted in Jordan. A survey of radon levels in Jordanian dwellings during the autumn season showed average radon levels of 57 Bq m$^{-3}$ [102]. More than 200 CR-39 detectors were used for the study of radon concentrations in the houses of Ajloun, a hilly town in the north of Jordan. The average radon concentration was 28 Bq m$^{-3}$ [103]. Another survey in some villages of the Ajloun district showed an average value of 36 Bq m$^{-3}$ [104].

### 3.17. Kazakhstan (I)

Surveys for radon concentration measurements began in the 1990s due to the awareness of potential health risks among workers and children. The results identified elevated

radon concentrations due to tectonic faults and geology (Kazakhstan is the world leader in uranium mining (https://world-nuclear.org/information-library/facts-and-figures/uranium-production-figures.aspx, accessed on 1 February 2023).

In one study, 23 measurements were conducted in the area covered by waste rock piles, as well as in selected private dwellings and gardens. The concentrations of radon ranged from 130 to 1200 Bq m$^{-3}$ indoors and from 20 to 90 Bq m$^{-3}$ outdoors. Indoor Rn concentrations were also measured in selected houses and public places near Ust-Kamenogorsk in U- and Th-rich areas. The values ranged from 22 to 2100 Bq m$^{-3}$, with a mean concentration (excluding the highest value of 2100 Bq m$^{-3}$) of 230 Bq m$^{-3}$ [71].

Several studies on radon (Rn) concentrations as well as radon decay products (RnDPs) in dwellings and workplaces have been described Kobal et al [52]. The authors reported that RnDPs are mostly below the limit of 200 Bq m$^{-3}$; however, in some mining regions, there are high levels of RnDPs, up to 37,000 Bq m$^{-3}$. Rn concentrations were measured in several places and the results also show high (with averages of 200–900 Bq m$^{-3}$) and extremely high Rn levels, up to 23,000 Bq m$^{-3}$ in the rooms of private houses.

Another extremely high radon concentration of 37,000 Bq m$^{-3}$ was found in one house in the city of Akchatau; this was due to the high radon exhalation rate from the wall, which reached up to 483 mBq m$^{-2}$ s$^{-1}$ [105].

### 3.18. Kuwait, Jordan, Syria, Yemen (V), and Arab Countries (Egypt, Libya, Tunisia)

Kuwait, Jordan, Syria, Egypt, Libya, Tunisia, and Yemen conducted regional radon surveys through a coordination research program (CRP) organized by the Arab Atomic Energy Agency (AAEA). The aim of the program was to create a database of indoor radon concentration levels in the region.

The results of the surveys showed that radon concentration levels in most of the dwellings averaged from 30 to 204 Bq m$^{-3}$, depending on the detector location (country), whilst in some old cities, and in an area close to a phosphate mine, the levels were found to be relatively high (>300 Bq m$^{-3}$) [106].

In the first study, 150 houses were investigated in the mid-1990s throughout the summer season. The PicoRad detectors (active charcoal) were exposed in each house for 48 h. The arithmetic mean was 14 Bq m$^{-3}$ with a range between 1 and 119 Bq m$^{-3}$ [107].

The same detectors (PicoRad) were used for a survey conducted from 2003 to 2005, employed in 300 houses in different parts of Kuwait. The average value for all locations was calculated at 33 Bq m$^{-3}$ with a range from 4 to 242 Bq m$^{-3}$ [108].

The first long-term survey of radon and thoron using SSNTD detectors was carried out in the period between 2015 and 2016. In total, 65 dwellings were checked. The results of the radon concentration measurements ranged from 10 to 202 Bq m$^{-3}$, whereas the thoron from the LLD (low limit of detection) was up to 35 Bq m$^{-3}$ [109].

### 3.19. Kyrgyzstan (I)

Small-scale studies to assess exposure to radon, thoron, and gamma radiation were conducted at several former uranium mining sites in Kyrgyzstan. Radon and thoron discrimination detectors were placed inside 67 private houses and public buildings. The results ranged from 10 to 2000 Bq m$^{-3}$ for radon and 3 to 800 Bq m$^{-3}$ for thoron, depending on the region and location of exposure. The authors concluded that, in general, there is no significant radiological risk to the general population residing in the area [110].

### 3.20. Korea (II)

The results of four indoor radon surveys in Korean dwellings were reported by Kim et al. [49]. Surveys were conducted in 1989, 1999–2000, 2002–2005, and 2008–2009. The main goal of these studies was to estimate the effective dose resulting from radon exposure for the general public. In all studies, SSNTD detectors were used to determine radon levels; in addition, in the third and fourth studies, thoron levels were measured. It has been reported that the mean radon concentration value during the first survey is higher

($104 \text{ Bq m}^{-3}$) than the others, i.e., 53, 66, and 79 $\text{Bq m}^{-3}$ for the second, third, and fourth surveys, respectively.

A regional survey in Korea involving 4670 dwellings revealed a log-normal distribution with a geometric mean of 94 $\text{Bq m}^{-3}$, with 6.6% of exceeding 300 $\text{Bq m}^{-3}$. An analysis of the results showed seasonal variation with the highest radon concentration in the winter and the lowest in the summer. In addition, dependence on the building construction style was observed [82].

A comprehensive study of indoor radon concentrations in South Korea was presented by Park et al. [111]. They analyzed 9271 published data from surveys conducted since 2011 and found that the population-weighted GM radon concentration for the entire country was 46 $\text{Bq m}^{-3}$. However, the unweighted results were estimated at 95 $\text{Bq m}^{-3}$ (AM) and 68 $\text{Bq m}^{-3}$ (GM), which is close to our AM result of 91 $\text{Bq m}^{-3}$, as presented in Table 4.

### 3.21. Lebanon (V)

In Lebanon, the population has one of the highest smoking rates in the world; when combined with radon, this poses a significant risk for lung cancer. Therefore, it seems reasonable to study the concentration of radon indoors. Radon concentrations were measured with SSNTD detectors in 26 houses for over 9 months (3 exposures, 3 months each) with results ranging from 20 to 343 $\text{Bq m}^{-3}$ [112]. Another study that was conducted in 24 places (indoor and outdoor) reported a log-normal distribution with median concentrations of 17 and 10 $\text{Bq m}^{-3}$, as well as ranges from 4 to 57 $\text{Bq m}^{-3}$ and 0.2 to 63 $\text{Bq m}^{-3}$ for indoors and outdoors, respectively [113].

### 3.22. Malaysia (III)

A large summary of radon measurements performed in Malaysia between 1994 and 2017 was presented by Ahmad et al [114]. They reported that indoor radon concentrations varied between minimum and maximum values from 11 to 3075 $\text{Bq m}^{-3}$.

### 3.23. Mongolia (II)

Indoor radon studies in concrete, brick, wooden, and Mongolian buildings were carried out over a period of 6 years to determine the concentration and annual dose of radon. As a result, the indoor radon concentrations were measured in various types of dwellings, i.e., concrete, brick, wood, Mongolian ger (Mongolian traditional dwelling); the average radon concentration was determined to be 26 $\text{Bq m}^{-3}$ [115].

### 3.24. Myanmar (III)

Short-term measurements that were conducted in several buildings using an active device (RAD7) showed a range of radon concentrations from 1 to 30 $\text{Bq m}^{-3}$ [116]. Another study was conducted in 50 1-story randomly selected residences in the municipality of Pabedan. Short-term measurements (2 h at each site) were made with the RAD7 device. The average value of radon concentration was 19 $\text{Bq m}^{-3}$ in the range of 3 to 84 $\text{Bq m}^{-3}$. The authors examined the relationship between the radon concentration and the floor type. Maximum concentrations were recorded in houses with bare concrete floors, and minimum concentrations with tiled flooring [117].

### 3.25. Nepal (IV)

Several studies on indoor radon concentrations have been conducted in Nepal in recent years. Track detectors (LR-115) were used to measure the radon concentrations in 41 randomly selected apartments in Kathmandu. The detectors were deployed for 100 days. The results showed an average value of 80 $\text{Bq m}^{-3}$ with a minimum of 8 $\text{Bq m}^{-3}$ and a maximum of 787 $\text{Bq m}^{-3}$ in one of the kitchens [118]. Another study was conducted in 50 randomly selected buildings located in 5 districts of Nepal. Moreover, in this case, passive detectors, specifically CR-39, were used. The minimum and maximum concentrations were <20 and 110 $\text{Bq m}^{-3}$, respectively [119]. Radon concentrations were assessed near a

waste management site located about 30 km from the city of Kathmandu. LR-115 detectors were installed in 17 homes for a period of 118 days. Radon concentrations ranged from 71 to 2026 Bq m$^{-3}$ [120].

### 3.26. New Zealand (Oceania)

In order to determine the annual radon dose and to check whether significant changes occurred since the previous national survey, a new survey of indoor radon concentration was conducted. A total of 260 houses were surveyed during the winter of 2015, with two measurements taken for each house. In this study, there was an average radon concentration of 23 Bq m$^{-3}$. An earlier survey conducted in 1998 by the National Radiation Laboratory examined 716 homes, revealing a concentration of 18 Bq m$^{-3}$. The authors of the report found no significant variations in radon concentration levels [121,122].

### 3.27. Oman (V)

The indoor radon concentration was estimated by collecting air on a filter and measuring radon progeny. The average radon concentration was calculated to be 21 Bq m$^{-3}$, but the number of locations has not been reported [123].

### 3.28. Pakistan (IV)

A large review of radon levels in Pakistan was provided by Matiullah and Wazir [124]. In summary, numerous studies have been conducted to measure radon levels indoors, outdoors, in workplaces, as well as in building materials. Most of these studies used integrated passive methods with SSNTD detectors (CR-39 and CN-85) and active methods (mainly the RAD7 device) to measure indoor radon and thoron concentrations. Many average values range from a few, i.e., 5 Bq m$^{-3}$, to several hundred, i.e., 800 Bq m$^{-3}$, and the average values are around 30–110 Bq m$^{-3}$.

### 3.29. Palestine (V)

The main objective of the study by Leghrouz et al. was to investigate indoor radon concentration measurements in Hebron province. They reported values that ranged from 23 to 580 Bq m$^{-3}$ with an arithmetic average of 91 Bq m$^{-3}$ [125]. Research on the measurements of radon concentrations and the seasonal fluctuations in dwellings during the winter and summer seasons was described in [126]. Indoor radon concentrations ranging from 30 to 655 Bq m$^{-3}$ and 35 to 984, in the summer and winter, respectively, were observed. The overall average in the summer (98 Bq m$^{-3}$) was lower than in the winter (124 Bq m$^{-3}$). The results of the measurements from 42 dwellings were described by Leghrouz et al. [127]. The lowest value (26 Bq m$^{-3}$) was observed in a living room and the highest (611 Bq m$^{-3}$) in a basement. The mean value was recorded as 118 Bq m$^{-3}$. Measurements of indoor radon concentrations in 46 dwellings were carried out by Abu-Samreh et al. [128]. The results ranged from 19 to 216 Bq m$^{-3}$ with a mean value of 79 Bq m$^{-3}$.

### 3.30. Papua New Guinea (Oceania)

Ten selected apartments in Lae, Papua New Guinea, were tested for radon and thoron levels. Measurements were conducted with passive and active detectors. Passive detectors with radon and thoron discrimination were exposed for three months in each location. In addition, a direct radon progeny sensor (DRPS) and direct thoron progeny sensor (DTPS) were used to measure the progeny of radon and thoron. An active device was used to validate long-term data. The reported radon concentration for the study area was in the range of 8–18 Bq m$^{-3}$ with an average value of 13 Bq m$^{-3}$, while thoron ranged from 1 to 4 Bq m$^{-3}$ with an average value of 3 Bq m$^{-3}$ [129].

### 3.31. Philippines (III)

The first nationwide survey in the Philippines was conducted in selected homes over a period of four years, from 1992 to 1995. Radon concentrations in rooms were measured

for six months using passive detectors based on CR-39 chips. In total, 2626 detectors were exposed. The indoor radon concentrations ranged from 1 to 58 Bq m$^{-3}$ with an average value of 21 Bq m$^{-3}$ [130].

### 3.32. Qatar (V)

Indoor radon concentrations in Qatar were carried out in select houses in various locations over one year using charcoal canisters. Results show that the mean indoor radon concentration in some dwellings of Doha city varies from 11 to 23 Bq m$^{-3}$ [131].

### 3.33. Saudi Arabia (V)

Indoor radon studies in Saudi Arabia began in the 1980s, but published data were limited; average radon concentrations were found to be 22 Bq m$^{-3}$ [132]. Moreover, 750 SSNTDs (based on CR-39 chips) were deployed to measure radon in apartments in Riyadh from October 2004 to June 2005. Results ranged from 2 to 69 Bq m$^{-3}$ with an average value of 18 Bq m$^{-3}$ [133].

In 2012, a research project was launched to build a national database on environmental radiation, and radon tests were carried out using passive detectors. The project measured radon levels in 786 dwellings and found concentrations of up to 195 Bq m$^{-3}$ with an average of around 25 Bq m$^{-3}$ [134]. Another study found that indoor radon concentrations ranged from 11 to 137 Bq m$^{-3}$, with an overall average of 32 Bq m$^{-3}$ for the 1119 dwellings surveyed [135].

On the other hand, there have been some small studies using active devices (mostly RAD7). For example, at Aljouf University, several offices were surveyed, and there was an average value of 12 Bq m$^{-3}$ [136]. Abuelhia [137] reported an average radon concentration level of 19 Bq m$^{-3}$ in apartments located in the city of Dammam (Eastern Province).

### 3.34. Singapore (III)

Screening measurements using an active device were conducted in a few residential houses in Singapore. In each location, the radon concentration was low, i.e., less than 15 Bq m$^{-3}$, except for one location with 80 Bq m$^{-3}$ [138].

### 3.35. Syria (V)

A national survey of indoor radon concentrations in Syrian homes was carried out in 1991–1993 using SSNTD detectors. The average radon concentration is given as 45 Bq m$^{-3}$. In a few houses (mainly in the southern region), radon concentrations were several times higher, which required remedial action. One of the aims of the study was to show whether there are differences in radon concentrations for different types of houses. In conclusion, it was shown that radon concentrations were higher in old mud houses with no tiling than in other buildings [139].

### 3.36. Tajikistan (I)

Radon concentrations in schools were measured by Muminov et al. [140]. The concentrations ranged from 42 to 331 Bq m$^{-3}$ with an average value of 98 Bq m$^{-3}$ on the first floor and 56 Bq m$^{-3}$ on the second floor. Another study was conducted in the Taboshar village to determine the radon concentrations in several homes and public buildings. Long-term (about 300 days) measurements of radon concentrations ranged from 15 to 330 Bq m$^{-3}$ with an average value of 75 Bq m$^{-3}$ [71].

### 3.37. Thailand (III)

Several indoor radon studies were conducted in Thailand, with arithmetic mean values between 16 and 109 Bq m$^{-3}$ [141,142].

A recent survey conducted in residential areas around old mines in southern Thailand showed elevated radon concentrations of up to 300 Bq m$^{-3}$. The authors concluded that

the study area was characterized by sedimentary and igneous rock formations, which may contribute to the high concentration of radon [81].

In the years 2018–2020, research was carried out to investigate the contribution of radon to the overall radiation dose. Indoor radon concentrations were measured using SSNTD detectors in 45 dwellings in Chiang Mai (7 districts). The results show statistical significance between the measured radon concentration in the burning season (63 Bq m$^{-3}$) and non-burning season (46 Bq m$^{-3}$). The average annual radon concentration was estimated at 55 Bq m$^{-3}$ [143].

The levels of radon, thoron, and their progeny near old mines and mineral deposit areas were studied by Rattanapongs et al. The reported radon concentrations in these areas ranged from 13 to 300 Bq m$^{-3}$, while thoron ranged from 115 to 184 Bq m$^{-3}$. The authors found that although elevated concentrations of radon and thoron were observed in the study area, they did not exceed the recommended values [81].

### 3.38. Turkey (V)

According to the results presented by Celebi et al. [144], a national survey to determine concentration levels in Turkish homes was carried out as part of the National Radon Monitoring Program. The main objective was to prepare a radon map of Turkey. Radon measurements were conducted in 7293 dwellings in 153 residential units of 81 provinces using SSNTD detectors. The results showed arithmetic and geometric mean concentrations of 81 Bq m$^{-3}$ and 57 Bq m$^{-3}$, respectively, with a geometric standard deviation of 2.3.

### 3.39. United Arab Emirates (V)

A regional study was conducted in Sharjah, United Arab Emirates (UAE), to measure indoor radon concentrations using an active radon detector. The results obtained from measurements in 61 houses in different parts of the city showed that radon concentrations ranged from 7 to 71 Bq m$^{-3}$ with an average value of 35 Bq m$^{-3}$ during the winter months. Radon levels were slightly lower in the summer [145].

### 3.40. Uzbekistan (I)

In Uzbekistan, the number of studies is limited, but several measurements were made in private homes and public places. One study used SSNTD to measure radon concentrations over a period of 3 to 4 months in the winter and spring. The radon concentrations ranged from 30 to 1050 Bq m$^{-3}$.

### 3.41. Vietnam (III)

In Vietnam, studies on radon levels in the environment began in the early 1990s. As reported by Dung et al., one of the first studies carried out in 1993 in Hanoi showed an average radon concentration of 27 Bq m$^{-3}$. Studies carried out to determine the background levels in houses at the site of a future nuclear power plant showed radon concentrations between 4 and 27 Bq m$^{-3}$, with an average of 11 Bq m$^{-3}$ [146].

In the vicinity of a coal mining area, the Rn concentration in dwellings was found to be 46 Bq m$^{-3}$ [147].

There are a number of measurement results reported in public places. For example, indoor radon concentrations were calculated based on data obtained from SSNTDs in university and school settings, ranging from 25 to 170 Bq m$^{-3}$ [148].

Nguyet et al. conducted a radon study to determine the dose received by visitors and staff at the Rong Cave located in Vietnam's Dong Van Karst Plateau Geopark. A ten-month measurement study showed radon and thoron variability depending on the season; average concentrations ranged from 206 to 6000 Bq m$^{-3}$ for radon and 74 to 546 Bq m$^{-3}$ for thoron [149].

It should be noted that although the authorities in Vietnam have not published official radiation safety standards for radon concentrations, there is a standard recommendation suggesting a radon concentration of 200 Bq m$^{-3}$ in buildings [150].

## 4. Identification of Problems

As previously shown in Table 4, the results of indoor radon measurements in some Asian countries are relatively old, dating back to the late 1980s and early 1990s. In this case, we can recognize three problems.

### 4.1. Bias Due to the Thoron Interference

The first problem involves the quality assurance of results due to old types of detectors, i.e., bare detectors, and/or thoron influence. The problem was widely discussed by Tokonami [151]. It was concluded that some old detectors have a high sensitivity to thoron, while others have a low sensitivity. It should be noted that the influence of thoron may be large if the detector is placed near the wall, even when low-sensitive detectors are used. As a consequence, the calculated annual dose can be overestimated.

### 4.2. Tendency toward "Green" Construction

The second problem is connected to the house construction. Modern technology and trends toward low, "green", energy houses can lead to tighter dwellings and reduced natural ventilation. Recent studies from Russia have shown an increasing trend of radon levels in buildings ranked with high energy efficiency indices [152]. In contrast to this, the results presented by McCarron [153] support the hypothesis that certified passive house buildings present lower radon levels.

### 4.3. Survey Design and Evaluation

In many papers reporting on the means of regional surveys, the 'sample representatives' issue is poorly (or not at all) discussed. Deviations from representative sampling can introduce biases in statistics, such as the mean, which renders the results questionable. See IAEA (2013), Section 3 of that report [154], and European Commission (2019), Section 2.4.5 [42] of that report, for further discussions of this very important subject. Moreover, the reporting of results that meet statistical standards is sometimes suboptimal, and uncertainty budgets are rarely addressed. In order to deliver reasonable results that can be internationally recognized, it is important to employ certified and QA-ed procedures, including calibration, sampling designs, individual measurements, and statistical evaluation. In many papers, QA is poorly reported.

## 5. Recent Developments

### 5.1. Thoron

In some regions of the world, thoron and its progeny contribute more to radiation doses than radon [84,155,156]. Kanse et al. [157] worked on developing a method that uses the exhalation rate of Tn from indoor surfaces as the basis for estimating the average concentration of Tn in indoor air. Taking this thoron concentration and appropriate conversion factors into account, the inhalation dose can be calculated.

### 5.2. Calibration Chambers

Karunakara et al. presented an innovative technique of using soil gas as a source of radon in a calibration chamber [158]. Constant radon concentrations in the range of 0.5–31 Bq m$^{-3}$ with a deviation of 5–15% were obtained by periodically injecting soil gas into the chamber. The time needed to obtain stable conditions is approximately 30–120 min depending on the required concentration.

As mentioned in Section 2, some Asian countries have (or will introduce) regulations on radon concentration levels in residential buildings and workplaces. For this purpose, measurements should be made and maintained; for the results to be reliable, the measurement systems must be checked and validated periodically. One method of maintaining quality is to conduct intercomparison tests. Janik et al. [159] presented the results of an experiment conducted in five radon and thoron measurement systems located in four Asian countries (China, India, Japan, and Thailand). They obtained good results when comparing

the radon systems (chambers). Deviations from the average concentrations did not exceed 5%. They also showed that the systems for testing and calibrating thoron devices still require further research.

### 5.3. New Detectors

One method of dividing radon detectors is to distinguish between passive detectors that integrate radon (e.g., SSNTD, active carbon, electret) and active detectors that measure radon continuously, based on, e.g., semiconductors, PIN photodiodes, etc. [160,161]

Although these detectors are used successfully, new methods are being developed and tested. One example involves a new detector presented by Hassanpur et al. [162]. They explored the possibility of alpha spectroscopy in detecting radon and its progeny using a microstrip gas detector. Experiment data were validated by using a MCNPX code and the spectrum from the microstrip detector was compared to the one obtained by the Atmos device. Results showed that the microstrip detector can measure radon and its progeny and it has the ability to extract the spectrum obtained from it.

Another gas-type detector, a micropattern gas detector (MPGD), was tested in order to measure radon and progeny [163].

Another example of the development of measurement techniques and methods is the system for measuring radon in the soil, as presented by Wang [164]. One of the challenges in measuring radon in soil is that moisture interferes with the results. The presented system attempts to avoid this problem by using a suitable waterproof membrane and a calculation algorithm.

### 5.4. Soil Radon as Tracer

A topic widely discussed in recent publications is the relationship between radon in soil and geohazard research. As a recent example, the purpose of the study presented by Ma et al. [165] was to show the mechanism that generates soil Rn anomalies by means of studying the geochemical behaviors of radionuclides in karst environments. They confirmed higher soil radon concentrations in karst compared to non-karst areas. They also found a significant positive correlation between Ra and $MnO_2$ ($R^2 = 0.86$), which implied that Ra mainly occurred in manganese oxide minerals.

### 5.5. Advanced Data Preprocessing and Evaluation

One important topic in the collection and interpretation of data, which mainly applies to active detectors, is imputation. Mir et al. presented a new imputation methodology (by feature importance) to generate an imputed dataset when dealing with soil gas radon concentration time series data. This approach provides more accurate mean value predictions [21].

A study by Rafique et al [166] investigated the complexity of radon, thoron, temperature, and a relative humidity time series via entropy, along with fractal dimension analysis techniques. Their results showed that the dependence and complexity of the time series data of soil gases are greater in the winter than in the summer.

### 5.6. Radon Awareness and Risk Communication

In addition to measuring radon and assessing the dose, increasing radon awareness and communication are very important tasks. Based on the survey results, it was concluded that the level of radon awareness among the people of Bahrain is low, with only 32.6% being aware of radon and its health hazards [167].

### 5.7. Radon Therapy

Currently, radon is being explored as an additional method of treatment for various diseases related to the respiratory system, pain, or rheumatism.

Review studies on radon therapy were compiled and presented by authors in Japan [168]. A comparison of the published research results shows that active oxygen in combination with radon gas has great potential in suppressing disorders and various types of diseases.

## 6. Discussion and Conclusions

### 6.1. Variation of Survey Density

Literature reviews of national and regional surveys published in peer-reviewed journals, as well as official reports from the USA [169] and Europe [42], show the number of radon measurements conducted per million inhabitants. Figure 1 shows the sampling density (sample size per population size) versus the human development index (HDI). Among Asian countries, on the one hand, Israel, which has a high HDI (0.919) and moderate radon concentrations (42–94 Bq m$^{-3}$), has the highest number of measurements; in Afghanistan, which has a low HDI (0.511) and a high radon concentration (>300 Bq m$^{-3}$), the number of measurements per million inhabitants is low. On the other hand, in India, China, and other Asian countries, new national surveys are in progress; therefore, the number of measurements will increase in the near future. However, due to the large population, this relationship will not dramatically improve.

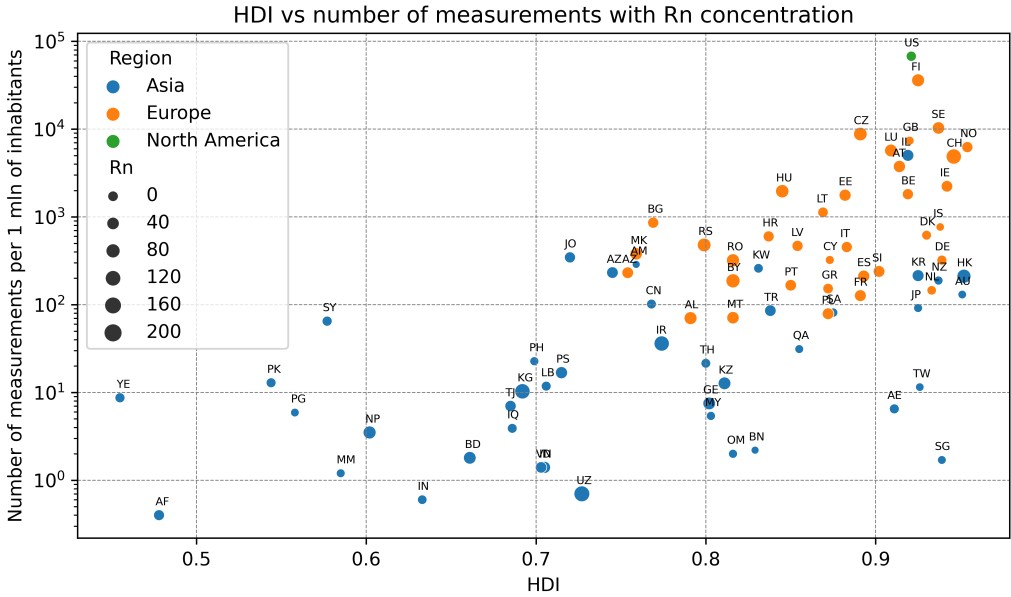

**Figure 1.** Measurements per million of inhabitants (log scale) versus the human development index (HDI). Countries are presented in Alpha-2 codes, as described in the international standard ISO 3166 [170]. European measurements refers only to ground floor rooms [42]. The presented average values calculated from scattered data should not be interpreted as the valid national average.

This review summarizes recent radon investigations implemented in Asian countries. As can be concluded from Figure 1, it is evident that the current radon surveying situation falls short when compared to the ones carried out in Europe and the USA. However, the number of projects related to radon and thoron, especially in India and China, has recently increased; we can expect more reliable results in the near future. It should also be noted that some countries have not shown any activity concerning radon policies, as far as can be concluded from the literature available to us.

### 6.2. Information Exchange and Logistical Harmonization

A platform for exchanging information within Asian countries should be created, similar to what exists in Europe, e.g., EURADOS WG3 (https://eurados.sckcen.be/working-groups/wg3-environmental-dosimetry, accessed on 1 February 2023), which deals with Environmental Dosimetry, or an information exchange and communication board, such as the

ERA (the European Radon Association, https://radoneurope.org/, accessed on 1 February 2023). One of the most important tasks will be to provide metrological support for the harmonization process of radon and thoron measurements in Asia, as well as for conducting periodic checks and intercomparison exercises, to ensure high-quality quality data and improve QA standards. At this moment, in Asian countries, international intercomparisons are organized ad hoc without coordination between countries. To date, none of the existing radon chambers in Asia has been accredited to the ISO 17025 standard, in contrast to Europe, where many scientific institutes and companies are accredited, e.g., BfS in Germany [171] and SÚJCHBO in the Czech Republic [172]. In Europe, the BSS, which must be transposed into national legislation by EU member states, and which requires establishing a national radon action plan, has proven to be a powerful incentive for developing a radon abatement policy, and has motivated many national and international research projects. A similar structure does not exist in Asia/Oceania, but a joint coordinated research policy would bring significant benefits.

*6.3. Radon Mapping and Geogenic Radon*

In addition, apart from level maps, a global Asian radon map with delineated radon priority areas (RPAs) could provide significant value for further discussions regarding the geographical distribution of individual and collective risks associated with radon exposure in Asia/Oceania, and for determining where resources—which are inevitably limited—should be allocated as part of a cost-effective radon abatement policy. Progress in mapping techniques is still ongoing, particularly regarding incorporating multiple predictors and addressing classification problems, such as the delineation of RPAs [173,174]. It might also be valuable for further direction in radon research, particularly for geohazard and earthquake studies, due to the high seismicity regions in countries such as Japan, the Philippines, Indonesia, Malaysia, Pakistan, China, and Taiwan (Figure 2).

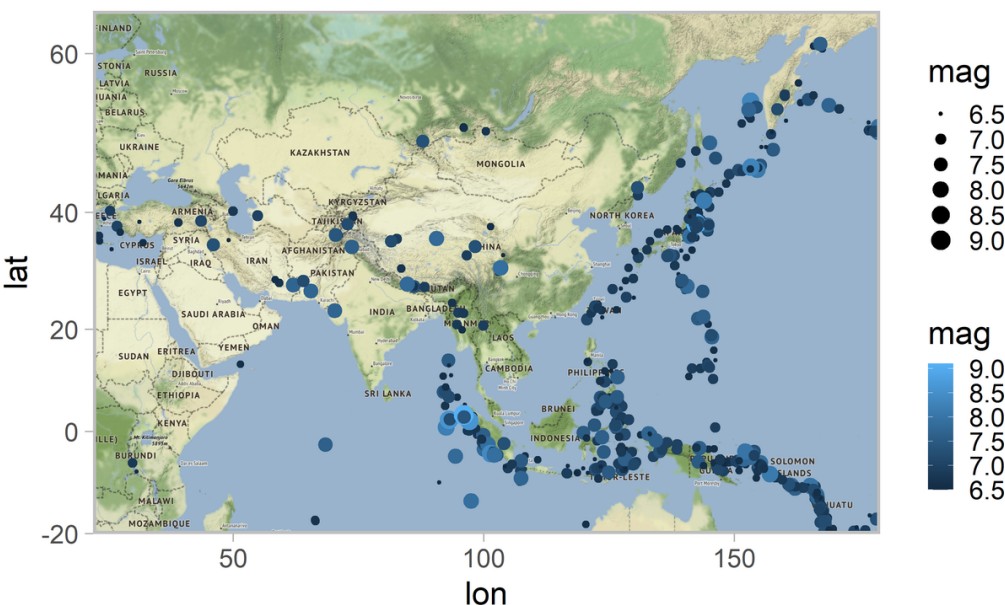

**Figure 2.** Geographical distribution of earthquakes with magnitudes higher than 6.5, between 2000 and 2022; source: USGS, US.

In principle, geogenic radon (mainly radon in soil gas or groundwater) can be used as a tracer for earthquake prediction. Many studies have been conducted on this subject, confirming the existence of the effects, on the one hand, and revealing a number of difficulties, on the other hand, which currently hinder its practical applicability. To date, there is no comprehensive procedure in earth sciences for predicting the time, location, and magnitude of an earthquake. One class of prediction methods relies on the analysis of

the spatial and temporal dynamics of Rn concentrations in soil air, groundwater, and the atmosphere [175,176]. In another class of proposed models, radon emission from the soil is believed to play a key role; the variation in atmospheric parameters (such as conductivity) could induce reactions in global electric circuits [177]. It should be mentioned that current research on machine learning techniques combined with advanced statistical methods is playing a promising role as a tool for detecting seismic anomalies in time series, e.g., due to earthquakes, as presented elsewhere [178]; however, research in this area is still in the early stages of development due to the lack of sufficient long-term measurement data.

To conclude, despite the many impressive efforts and achievements in radon policy in Asia/Oceania, many tasks still need to be done. To a large degree, this can be attributed to the wide diversity and heterogeneity of the region in many respects, as well as the need for further scientific research to be conducted.

**Author Contributions:** Conceptualization, M.J; Methodology, M.J. and P.B.; Formal analysis, M.J., P.B. and M.M.H.; Investigation, M.J., P.B. and G.C.; Writing–original draft, M.J.; Writing–review & editing, G.C. and M.M.H.; Visualization, M.J. All authors have read and agreed to the published version of the manuscript.

**Funding:** This research received no external funding.

**Institutional Review Board Statement:** Not applicable.

**Informed Consent Statement:** Not applicable.

**Data Availability Statement:** The data included in this work came from published articles or reports. Links to the sources are in the references.

**Acknowledgments:** The authors wish to acknowledge the G.I.4 Unit of the Joint Research Centre of the European Commission for the development of the European Atlas of Natural Radiation and, in particular, for granting access to statistical data from the European Indoor Radon Map.

**Conflicts of Interest:** The authors declare no conflict of interest.

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
