# Peer review of "Indoor Radon Research in the Asia-Pacific Region"

_atmosphere, doi:10.3390/atmos14060948_

Round 1

Reviewer 1 Report

The present article represents an important review of studies on radon in the Asian-Pacific region. It represents an important contribution in the field of radon, offering a succinct presentation of the studies carried out both nationally and regionally in the mentioned region. I would have one suggestion, in the head of Table 4, next to the radon concentration, the unit of measure should be mentioned! Congratulations to the authors for their effort and work!

Author Response

Thank you for your comment. Head of Table 4 is corrected.

Reviewer 2 Report

The article contain a review on experimental data regarding Radon indoor in vast area of Asia and Oceania. This concern to a areas on which majority of human world population live.  It is area of very different character of human dwellings, with extreme large modern  cities and small villages, with extremely different climate. Continental  cold winters in majority of Asia and tropical areas of Oceania. The construction of houses and therefore resulting possible radon accumulation inside happens. So, taking this all within a single paper, not a book, is a hard task. I think it is well written and I recommend it for publication with only small corrections.

Table 4 . In caption it should be said that those are in door values. However, what is a justification for a single value of 84 Bq/m3  for Iran, where on previous page a paper [14] with a negative  correlation between  radon concentration and cancer mortality was observed. For some other countries a range is given instead of a single average, so it should be done also for all countries where a range is available, and especially for country with high spread like Iran. The Iran, as a country with small but extreme high level area existing, should not be represented by a mean value. Especially, that later in page 9 very high values are given. What is a difference between “State of Palestine” and “Palestine”?

India – are those results contain also that of Karela? Are there any data for indoor Rn from high natural radioactivity areas available?

Author Response

Thank you very much for your valuable comment.

In some studies, the authors did not report the mean values of their studies, so it is impossible to calculate the average value without knowing the distribution of radon during the studies. The average has been calculated by the authors based on the averages obtained from the studies and the availability of data. Note that in this paper we are not providing values for policy makers, but rather as an overview for the community to see if there is a radon problem and if more effort needs to be made in a particular region/country.

 “State of Palestine” and “Palestine” are the same regions. There is a mistyping. It is corrected now, with average and max value.

 We included results from Kerala.

 We added more data from Iran.

Reviewer 3 Report

This manuscript is a very significant summary of radon research in Asia. Actually, it only describes indoor measurements and does not include, for example, soil gas measurements, as the title indicates. Perhaps changing the title (Indoor radon research...) might be reasonable. 

Line 65: You wrote, that section 2 is about the regulation in Asian and Oceanic countries, but from Line 73 until Line 89 you discussed the regulation in the other part of the world. This part is more than half part of section 2. 

Why are the surveys separated into national and regional surveys? What is the difference between them, because the descriptions do not show it.

Author Response

Thank you very much for your comments.

The title of the manuscript is corrected.

Lines 66-67: The following sentence has been added: “..and compare them with regulations from other regions of the world, especially the EU and North America”

We decided to resign from the division into nationwide and regional studies. A more detailed explanation can be found in the text, paragraph 3, lines 96-127.